# Surface Morphology and Spectroscopic Features of Homoepitaxial Diamond Films Prepared by MWPACVD at High CH_4_ Concentrations

**DOI:** 10.3390/ma15217416

**Published:** 2022-10-22

**Authors:** Javier Sierra Gómez, José Vieira, Mariana Amorim Fraga, Evaldo Jose Corat, Vladimir Jesus Trava-Airoldi

**Affiliations:** 1Associated Laboratory of Materials and Sensors, National Institute for Space Research, São José dos Campos 12227-010, Brazil; 2School of Engineering, Mackenzie Presbyterian University, São Paulo 01302-907, Brazil

**Keywords:** single crystal diamond, microwave plasma-assisted CVD, morphology, defects, homoepitaxial growth

## Abstract

Single crystal diamond (SCD) is a promising material to satisfy emerging requirements of high-demand fields, such as microelectronics, beta batteries and wide-spectrum optical communication systems, due to its excellent optical characteristics, elevated breakdown voltage, high hardness and superior thermal conductivity. For such applications, it is essential to study the optically active defects in as-grown diamonds, namely three-dimensional defects (such as stacking faults and dislocations) and the inherent defects arising from the cultivation method. This paper reports the growth of SCD films on a commercial HPHT single-crystal diamond seed substrate using a 2.45 GHz microwave plasma-assisted chemical vapor deposition (MWPACVD) technique by varying the methane (CH_4_) gas concentration from 6 to 12%, keeping the other parameters constant. The influence of the CH_4_ concentration on the properties, such as structural quality, morphology and thickness, of the highly oriented SCD films in the crystalline plane (004) was investigated and compared with those on the diamond substrate surface. The SCD film thickness is dependent on the CH_4_ concentration, and a high growth rate of up to 27 µm/h can be reached. Raman spectroscopy, high-resolution X-ray diffractometry (HRXRD), scanning electron microscopy (SEM), surface profilometry and optical microscopic analyses showed that the produced homoepitaxial SCD films are of good quality with few macroscopic defects.

## 1. Introduction

The homoepitaxial and heteroepitaxial growth of diamonds have been explored and optimized for different technological applications [1]. In recent years, many advances have been made to enable the growth of high-quality homoepitaxial single-crystal diamond (SCD) films for the development of optical and optoelectronic devices, as SCD films are excellent transparent semiconductor materials that also exhibit exceptional electrical, thermal, chemical and mechanical properties [2]. This has stimulated the research and development of diamond-based semiconductor devices with potential use in harsh environments such as high temperatures (>400 °C), high voltages (>10 kV) and extreme radiation conditions [2]. However, the wide adoption of SCD in microelectronics and optoelectronics has still been limited by the size of available high-quality natural and synthetic HPHT diamond crystals [3].

Different CVD growth techniques and conditions have been investigated and used to produce high-quality SCD films [4,5,6,7,8,9]. The effects of temperature, pressure and holder geometry on the homoepitaxial SCD growth by microwave-assisted chemical vapor deposition were discussed by Widmann et al. [4]. Bushuev et al. showed that an SCD growth rate of up to 82 μm/h can be achieved by optimizing the substrate temperature [8]. More recently, Vikharev et al. reported the growth of high-quality (100)-oriented SCD with a growth rate of 6−8 μm/h [5]. In another recent article, Zhang et al. reported the effect of CH_4_/H_2_ flow rates varied from 6% to 6.2%, 6.4%, 7% and 8% on the growth of SCDs. It was observed that under the optimized condition of 6.4% methane concentration, the growth rate was 3 µm/h [9]. The influence of methane concentration on the growth rate and structure of single-crystal CVD diamond has also been discussed in other previous studies [10,11,12]. Achard et al. investigated the evolution of the growth rate varying the CH_4_ concentration from 2% to 7% [11], whereas Bogdan et al. used from 6% to 15% [12].

Although progress has been made, an improved understanding of high-quality CVD SCD film large area growth with suitable thickness is still required to design and fabricate diamond-based semiconductor devices with high performance and good reliability. In this paper, we exploit the MWPACVD technique to obtain a very high growth rate of SCD films by varying the methane concentration from 6% to 12%. A very high SCD film growth rate of up to 27 µm/h was obtained. In addition, the structural quality of the films is discussed in detail.

## 2. Materials and Methods

The experiments reported here have been performed on <100> oriented yellow HPHT type Ib seed substrates (3 × 3 × 1.1 mm^3^) from the Chenguang Machinery & Equipment Co., Ltd. (Changsha, China). The full width at half maximum (FWHM) of the Raman diamond spectral peak and HRXRD rocking curve were 4.8–5.3 cm^−1^ (1333.2 cm^−1^) and 0.012°, respectively. By optical profilometry in VSI measurement mode., all selected substrates presented an arithmetic average roughness (Ra) lower than 7.0 nm, for an analyzed area of 300 × 230 μm^2^. The substrates were first cleaned in effervescent aqua regia acid solution at 100 °C for 60 min, followed by washing with acetone, isopropyl alcohol and deionized water in an ultrasonic bath for 10 min and then dried with a heat gun. This was done with the aim of removing traces of metallic elements, graphite and other organic contaminants. Each sample was loaded individually on a modified substrate carrier into the chamber, which was then evacuated with a diffuser pump to 10^−5^ Torr for 30 min.

A feed gas mixture of hydrogen and methane gases with purity levels of 99.999% and 99.99%, respectively, was used for all experiments. The methane (CH_4_) concentration in the gas phase was varied from 6% to 12% in balanced hydrogen. Prior to film growth, the substrates were etched in a H_2_ plasma for 40 min at 1040 °C. SCD films were grown in a cylindrical cavity MWPACVD reactor designed to operate with a 6 kW and 2.45 GHz microwave generation system, specifically optimized for this experiment with the parameters given below. All samples were exposed to the same growth environment with 3.6 kW microwave power, a pressure of 150 Torr and a temperature of 1060 ± 10 °C for a total growth time of 10 h. The methane concentration was 6%, 8%, 10% and 12% diluted in H_2_ at a total flow of 200 sccm, as shown in Table 1. This CH_4_ concentration range was chosen based on the conclusions of previous studies on CVD single-diamond growth without nitrogen addition [9,13]. Temperature control was performed by micrometrical positioning of a copper heat exchanger with a servomotor and monitored with a high resolution (0.1 °C) two-color infrared pyrometer (MERGENTHALER LASCON 101 LPC-03, Neu-Ulm, Germany).

After growth, samples had their bottom faces cleaned by simple grinding with a polishing cloth and 3 µm diamond paste. The samples were then boiled again in aqua regia solution at 100 °C for 60 min, rinsed in deionized water, and cleaned in an ultrasonic bath for 10 min with acetone, followed by further ultrasonic cleaning in isopropanol for 10 min. The SCD films’ structural quality was investigated by: (i) Raman and photoluminescence (PL) spectroscopy using a Horiba Labram HR evolution spectrometer (Kyoto, Japan) applying a 514.5 nm wavelength Nd:YAG laser source at room temperature, (ii) digital camera images using a Digilab DI-106T stereoscope (São Paulo, Brazil), (iii) SEM micrography using Tescan VEGA3 and MIRA3 electron microscopes (Brno, The Czech Republic) to analyze surface morphology, (iv) optical profilometry using a Veeco WYKO NT1100 profilometer, and (v) high-resolution X-ray diffractometry (HRXRD) using a PANalytical X’Pert setup (Malvern, UK) equipped with a four-crystal Ge(220) monochromator placed after the copper X-ray tube, which leads to Cu Kα_1_ radiation (λ = 1.5406 Å) for the incident beam, and an open gas proportional detector with an acceptance angle of 1° for the diffracted beam.

## 3. Results

### 3.1. Raman Analysis

The whole surface of the grown SCD films was analyzed by Raman spectroscopy to evaluate the homogeneity of structural quality. A value of FWHM of the characteristic diamond peak centered at 1332 cm^−1^ was used. The lowest FWHM obtained for each grown sample was compiled for comparison (Figure 1).

The structural quality of the substrates was also evaluated by this methodology. This allowed us to create an initial standard to compare each sample result separately. In Figure 2, values for the FWHM of the Raman Scattering diamond peak are summarized for each sample in relation to the position of the measurement at the sample surface.

### 3.2. PL Measurements

It is also possible to identify some features of the grown film when a PL spectrum is measured at the sample and compared to the substrate, as shown in Figure 3.

Luminescence is present at discrete wavelengths accompanied by an almost linear increase in the spectrum baseline, which is a common feature in polycrystalline CVD diamonds. This phenomenon is related to the incorporation of non-diamond phases in the grown material, which may be influenced by the growth rate of the films. If the crystal grows too quickly, some hydrocarbon radicals from CH_4_ dissociation in the surface do not properly react with the exposed binding sites and are incorporated with different hybridization to the lattice. Therefore, a delicate balance between diamond growth rate, and CH_4_ and nitrogen concentration is critical to achieving high quality, color grade and application properties.

### 3.3. Growth Rate

As Figure 4 shows, there is a significant increase in the thickness of grown films as the CH_4_ concentration is increased. All samples were grown for the same time of 10 h. The first sample grown at 6% CH_4_ concentration achieved a thickness of around 165 µm and the thickness of the last one grown at 12% CH_4_ concentration reached around 270 µm. This difference in deposited material could have positively impacted the measured FWHM. In addition, accelerated growth may cause luminescence increases, once more non-diamond particles are incorporated into the film, whereas a slight improvement in diamond quality is also perceivable with the increase in CH_4_ concentration.

The growth rate is also impacted by nitrogen addition and its effects have been extensively studied by many different authors [14,15]. In our experiments, no intentional nitrogen was added to the process, but it was present in feed gases as a contaminant, and also introduced through leakages in the vacuum system. The total amount of nitrogen added to the process was estimated using equation 1 for the amount of nitrogen entering the chamber from vacuum leakage:(1)$$\left[N_{2}\right]=\left[0.78\left(\frac{\Delta P\frac{V_{c}}{760}}{\dot{m}}\right)\right]10^{6}$$
where [*N*_2_] is the amount of nitrogen entering the chamber through vacuum leakages in ppm, Δ*P* is the variation of pressure per minute in Torr, *V_c_* is the chamber volume in cm^3^, *ṁ* is the total gas flow in sccm, and 0.78 is the ratio of nitrogen content in air. By this method, we calculated that the amount of nitrogen from leakage was 46.6 ppm. For the nitrogen coming from gas impurities, we applied Equation (2) based on information provided by technical analysis reports from the gases’ manufacturers:(2)$$\left[N_{2}\right]=\left[\left(1-\left[\frac{N_{2}}{H_{2}}\right]\right)\dot{m}H_{2}+\left(1-\left[\frac{N_{2}}{CH_{4}}\right]\right)\dot{m}CH_{4}\right]10^{6}$$
where [*N*_2_] is the total amount of nitrogen in ppm entering the process from feed gases, [*N*_2_/*H*_2_] and [*N*_2_/*CH*_4_] are the concentrations of nitrogen in ppm in the hydrogen and methane gases respectively, and *ṁ**H*_2_ and *ṁ**CH*_4_ are the total flow of hydrogen and methane in sccm. The results considering the impurity levels of the gases we used are summarized in Table 2 [16].

In a recent article, Wu et al. discussed the influence of nitrogen addition on the defects, morphologies and growth rate of single-crystal CVD diamond films produced by microwave plasma chemical vapor deposition (MPCVD) [17]. The CH_4_/H_2_ ratio was set at 4% with a hydrogen flow rate of 200 sccm. The nitrogen concentration in the CH_4_/H_2_/N_2_ gas mixtures was varied from 0–9524 ppm. The highest growth rate achieved was 23 µm/h at a N_2_ concentration of 5261 ppm, whereas in our study without intentional nitrogen addition we obtained a growth rate of 27 µm/h. Furthermore, according to their conclusions, as the nitrogen concentration increases, the structure and content of the defects evolve as well, with the domination of point defects, probably leading to lattice deformation in the as-grown diamond films in addition to eventually leading to variations in the surface roughness and uniformity of the diamond films.

### 3.4. Surface Morphology

Images obtained with optical microscopy are shown in Figure 5. It is possible to observe that our grown layers are nearly colorless over the yellow, type Ib HPHT seed substrate, despite the considerable amount of nitrogen incorporated in the films. For the sample grown at a CH_4_ concentration of 8%, shown in Figure 5b, some of the striations from step-growth are visible, this feature is present in all samples and will be further examined in the discussion of the results of the SEM and profilometry analyses below.

SEM analyses were employed to identify in detail some features of the striations formed by the step-growth regime of the films. It is noticeable in the obtained micrographs (Figure 6), that as well as the formation of steps, some square-shaped features are present (Figure 6a–c). This is related to the growth of the steps in a screw pattern centered at dislocations present in the crystal [18].

In Figure 7, we show results for profilometry conducted to better understand how growth steps can influence the morphology of the top surface. We found that surface roughness can be significantly affected by CH_4_ concentration, with the length of the growth steps being inversely proportional to the increase in CH_4_ concentration. This caused the roughness (Ra) to fall from 629.13 nm for the sample grown with 6% CH_4_ to 171.59 nm for the sample grown with 12% CH_4_. Values for Ra of all samples are summarized in Table 3.

### 3.5. High-Resolution X-ray Diffraction Analysis

X-ray diffraction curves (ω scans) were measured around the (004) Bragg peak for all samples. Figure 8 shows these curves for the samples with the different methane concentrations investigated here. To correct the offset in the ω axis, all curves were shifted to the central position to allow comparison. The X-ray curves are very similar and independent of the CH_4_ concentration. Their FWHM values, displayed in Table 4, range from 0.014° to 0.028° and are equivalent to values published in the literature [19,20]. It is important to observe that the attenuation length of the X-ray incident beam in diamond crystals calculated for λ = 1.5406 Å is 570 μm, which is three times the thickness of the grown diamond films. Therefore, the measured X-ray curves are a convolution of the diffracted beam coming from both the diamond film and the diamond substrate. This result demonstrates that the diamond films grown in this work present a crystal quality comparable to the commercial HPHT substrate. The (004) diamond Bragg peak was calculated by the dynamical theory of X-ray diffraction using the Takagi–Taupin equations [21]. For comparison, the calculated curve is also included in Figure 8, and its linewidth is found to be 0.001^o^, 20 times smaller than the measured linewidths.

## 4. Discussion

Nitrogen-vacancy defect complexes are identifiable in the spectra by the visible bands centered at 575 nm and 637 nm in the PL spectra, for NV^0^ and NV^-^, respectively. At the same time, the substrate did not present the same features, even though it is a type Ib substrate that also has nitrogen incorporated in the grown diamond. This difference in the incorporation of nitrogen, as well as its concentration, inside the diamond lattice, plays a key role in many properties of the film, especially in its color. CVD-grown diamond usually presents a brown hue, which can become darker with higher nitrogen concentrations in the gas phase. This color is attributed to the color centers of the vacancies. HPHT-grown diamond has nitrogen as single substitutional atoms in the lattice, and this kind of incorporation leads to the characteristic yellow color of these synthetic stones [22].

An additional band that is present in some of our samples is the 737 nm SiV center. This is a commonly-observed band for CVD diamond, as microwave plasma reactors usually employ solutions to handle the magnetic and electric fields involving quartz domes, tubes, or windows. Silicon from these components is removed by plasma etching or heating and can incorporate into the diamond lattice as a substitutional atom, creating neighboring vacancies in the structure. The emission from this center is related to the nitrogen content in the grown diamond. Tallaire et al. observed that higher levels of nitrogen doping increase the intensity of this band [15].

Our samples also presented significant lateral growth of the same order as the top surface growth. This is a common feature in MWPACVD diamond homoepitaxial layers, also described by other authors [23,24]. The lateral growth can be observed by the black line at the borders of the samples. The thickness of these borders is roughly the film thickness itself.

In a general manner, this growth pattern is observed for single and polycrystalline diamonds oriented in the [001] direction [18,25]. Tallaire et al. also reported the contribution of nitrogen content to the formation of growth steps and observed a surface free of this surface feature for a film grown with no added nitrogen [15]. However, it is not clear if the addition of nitrogen per se causes a reduction of the step size or if this could only be related to the growth rate. A more detailed study is needed to elucidate the phenomena involved in their formation.

## 5. Conclusions

Homoepitaxial single crystal diamond (SCD) films were produced using the MWPACVD process with a high deposition rate from 16.5 to 27 µm/h by varying the methane concentration in the reactor from 6% to 12%. The thickness of the SCD films was correlated with their structure, morphology and appearance. The results obtained demonstrate the good structural quality of the grown SCD films. This study represents an advance in attempts to optimize the growth of high-quality SCD films for different applications. It was possible to evaluate the influence of the concentration of CH_4_ by scanning electron microscopy, which was confirmed by optical profilometry, with a decrease in the surface roughness from Ra = 624.13 nm with 6% CH_4_ to Ra = 171.59 nm with 12% CH_4_. Thus, Ra was inversely proportional to the percentage of CH_4_ in the gas mixture. Therefore, the smoothest step growth surface was obtained with 12% CH_4_; which was, however, impacted by the higher unintentional incorporation of nitrogen into the film from the CH_4_ gas. In terms of structural quality, 8% CH_4_ was the best condition with an average FWHM of 4.6 cm^−1^. At the same time, in terms of the crystalline quality of the grown SCD, the X-ray curves were very similar and the FWHM varied between 0.014–0.018°, except for 10% CH_4_, which was 0.028°; and the PL measurements indicated that this concentration of CH_4_ favored the formation of NV defects. These results showed that diamond films grown at 6–12% CH_4_ had good crystalline quality. Overall, this study contributes toward the understanding of single CVD diamond growth and their morphology and structure.

## Figures and Tables

**Figure 1 materials-15-07416-f001:**
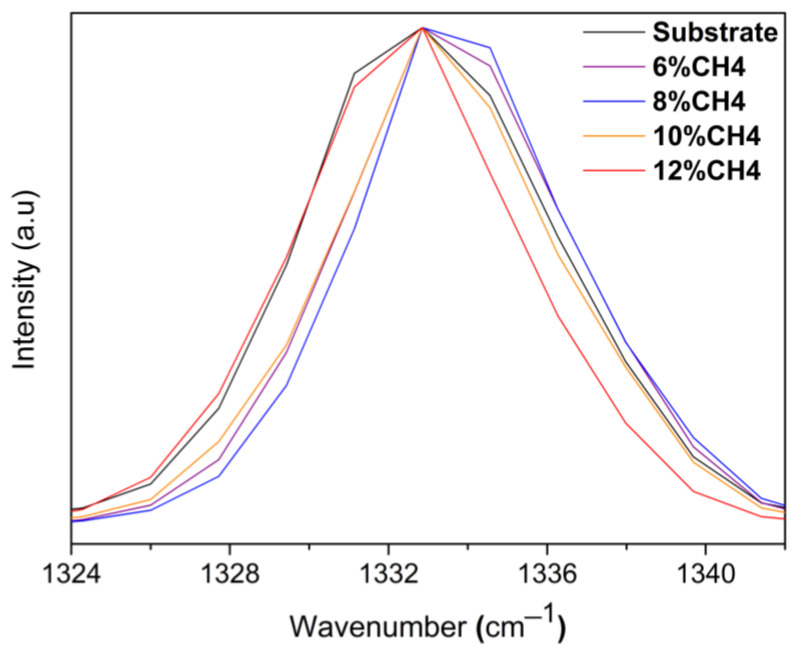
Raman spectra of as-grown MWPACVD diamond films on top of a diamond single-crystal seed substrate by varying CH_4_ gas concentration of 2% to 12%.

**Figure 2 materials-15-07416-f002:**
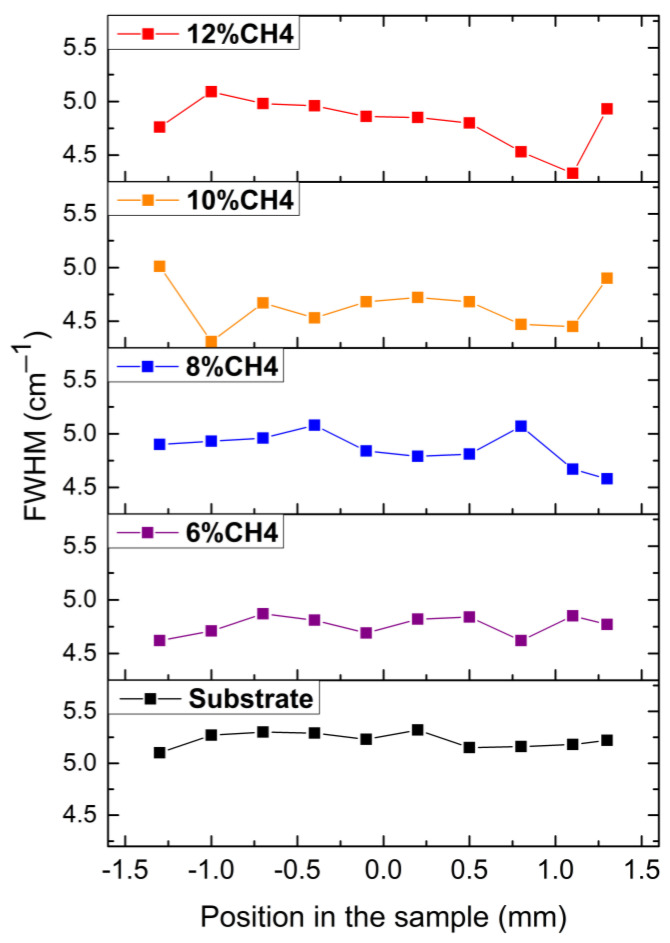
FWHM of the Raman spectra of diamond films grown under different CH_4_ gas concentrations as a function of the position at the sample surface.

**Figure 3 materials-15-07416-f003:**
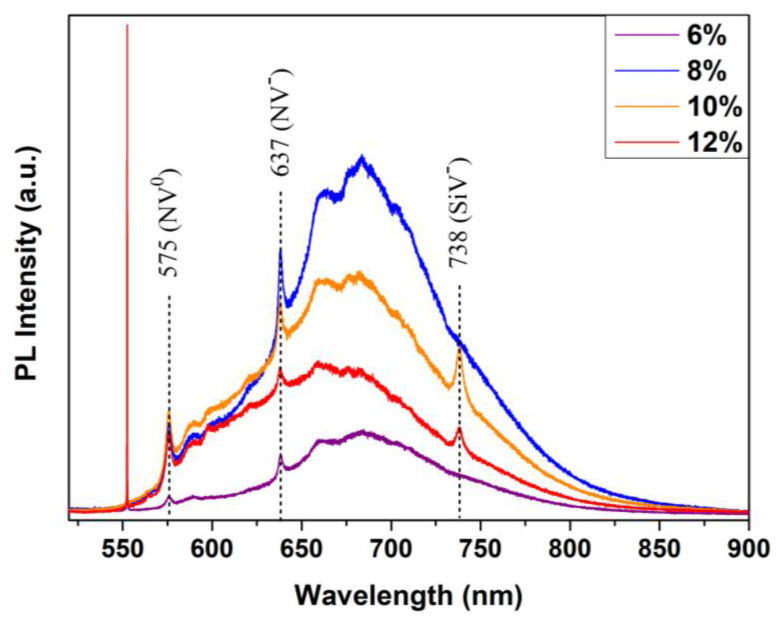
PL spectra at room temperature of the diamond films grown at 6%, 8%,10% and 12% CH_4_ in the deposition process.

**Figure 4 materials-15-07416-f004:**
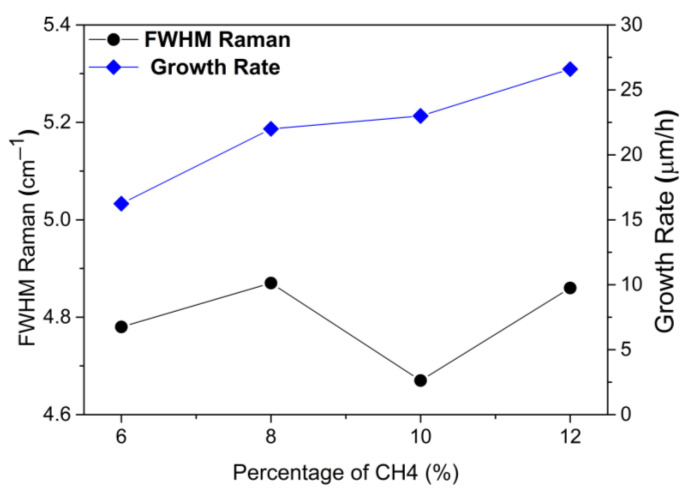
Growth rate and FWHM of the Raman diamond peak as a function of CH_4_ concentration.

**Figure 5 materials-15-07416-f005:**
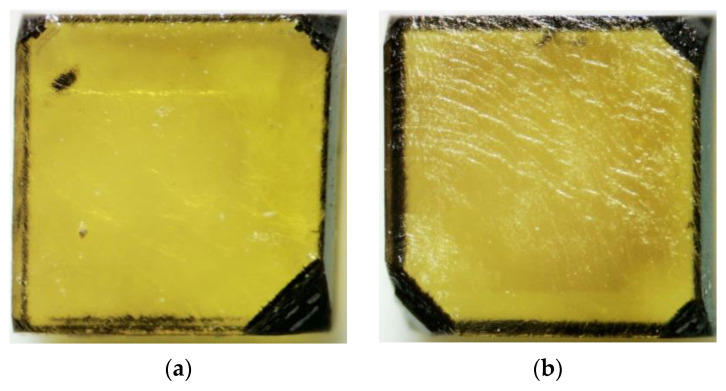
Optical images of as-grown MWPACVD diamonds on top of HPHT type Ib seed substrates with different CH_4_ gas concentration: (**a**) 6%, (**b**) 8%, (**c**) 10%, and (**d**) 12%.

**Figure 6 materials-15-07416-f006:**
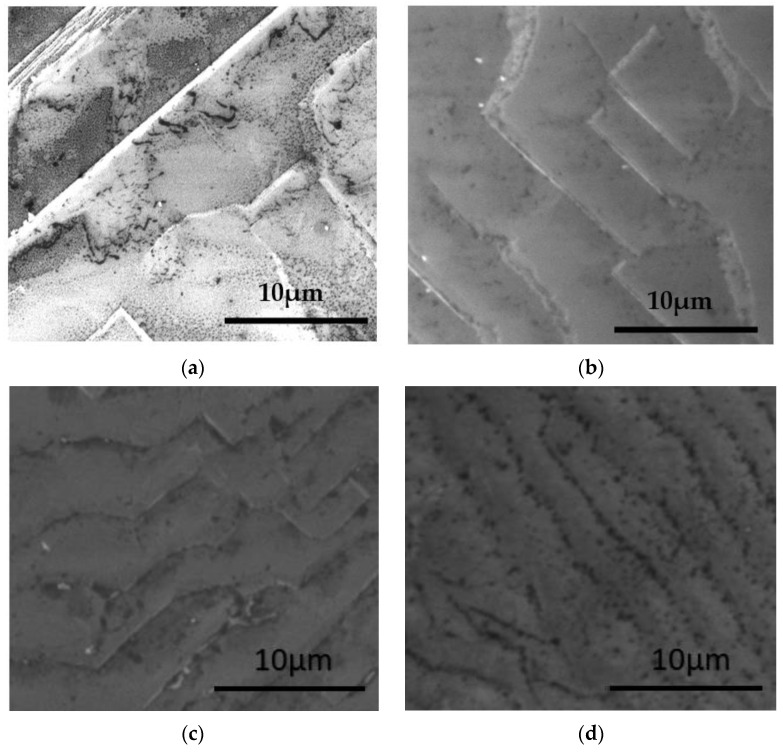
FEG-SEM Images of as-grown MWPACVD diamond films on top of diamond single-crystal seed substrates with different CH_4_ gas concentration: (**a**) 6%, (**b**) 8%, (**c**) 10%, and (**d**) 12%.

**Figure 7 materials-15-07416-f007:**
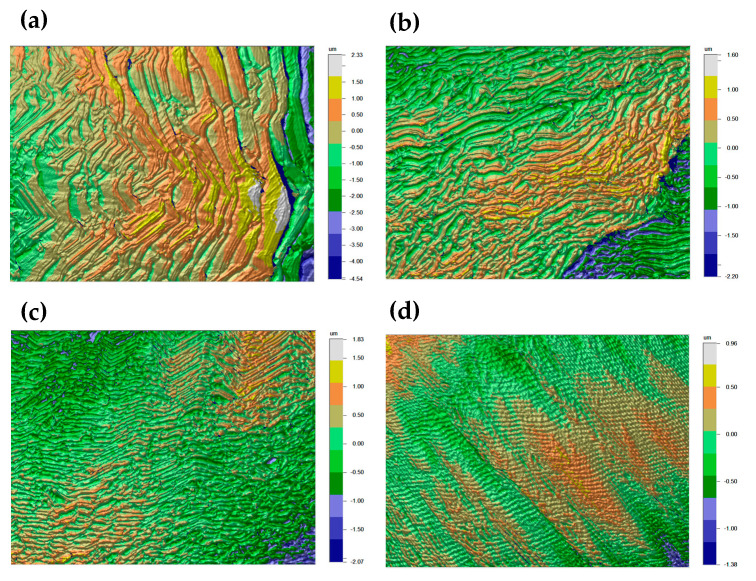
Profilometry of the surface of samples grown with (**a**) 6% (**b**) 8% (**c**) 10%, and (**d**) 12% CH_4_.

**Figure 8 materials-15-07416-f008:**
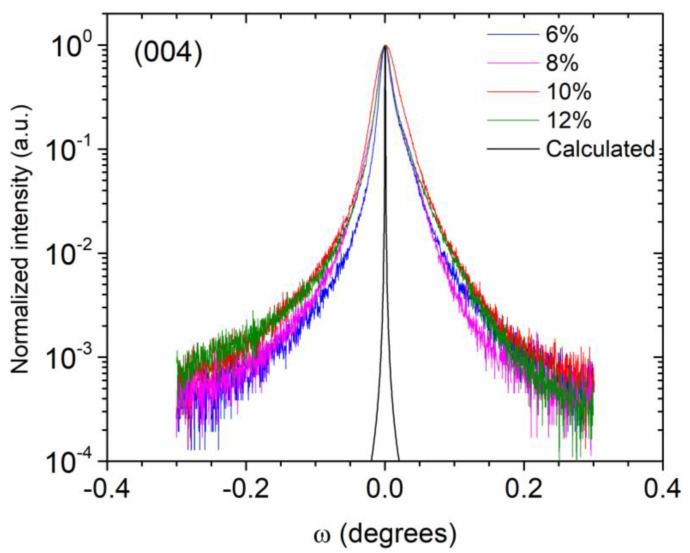
HRXRD rocking curve (ω scan) around the diamond (004) reflection of all samples.

**Table 1 materials-15-07416-t001:** SCD growth parameters for different CH_4_ concentrations.

Sample	%CH_4_	Temperature (°C)	Pressure (Torr)	Power (kW)	Rate Growth (µm/h)
SCD-06	6	1060	150	3.62	16.2
SCD-08	8				22
SCD-10	10				23
SCD-12	12				26.6

**Table 2 materials-15-07416-t002:** Thickness measurements for each CH_4_ and N_2_ from feed gases concentrations.

[*CH*_4_/*H*_2_] (%)	Thickness (µm)	[*N*_2_] (ppm)
6	165	6
8	215	8
10	226	10
12	270	12

**Table 3 materials-15-07416-t003:** Collected data of surface Ra for each sample.

[*CH*_4_] (%)	Ra (nm)
6	629.13
8	396.24
10	353.01
12	171.59

**Table 4 materials-15-07416-t004:** FWHM of the samples.

[*CH*_4_] (%)	FWHM (°)
6	0.014
8	0.018
10	0.028
12	0.018

## Data Availability

Not applicable.

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
