# Peer review of "Surface Morphology and Spectroscopic Features of Homoepitaxial Diamond Films Prepared by MWPACVD at High CH4 Concentrations"

_materials, 2022, doi:10.3390/ma15217416_

Round 1

Reviewer 1 Report

In this paper, the growth of SCD films on a commercial HPHT substrate of single-crystalline diamond using the microwave plasma chemical vapor deposition (MWPACVD) method at a frequency of 2.45 GHz with a methane (CH4) concentration variation from 6 to 12% while keeping other parameters constant was investigated. The effect of CH4 concentration on the properties, such as structural perfection, morphology, and thickness, of highly oriented SCD films in the crystal plane (004) was investigated and compared with films on the surface of a diamond substrate. It is shown that the thickness of the SCD film depends on the CH4 concentration and a high growth rate of 27 μm/h can be achieved. Raman spectroscopy, high resolution X-ray diffractometry (HRXRD), scanning electron microscopy (SEM), surface profilometry, and optical microscopic analysis were used to investigate the structural perfection of homoepitaxial SCD films grown at different methane concentrations.

In general, the article is of an ordinal nature and presents standard descriptive results. For application in microelectronics today it is necessary to form synthetic diamonds on substrates with a diameter of 200-300 mm.

The only positive aspect of the work is that the effect of methane concentration on the growth rate of the monocrystalline diamond film has been demonstrated.

There are a number of comments to be made on the article:

(1)    Figure 4 shows the dependence of the growth rate on the concentration of methane. At a concentration of 10%, a slowing of the film growth rate and a decrease in the half-width of the Raman spectrum curve are observed. Also in Figure 8 and Table 2, there is a sharp increase in the half-width of the rocking curve at a methane concentration of 10%. It turns out that this point is either critical or an artifact. In this case, it is advisable to conduct a study in the methane concentration range of 6-12% in increments of 1%, which will allow a more detailed study of the growth process. In Figure 1, it is necessary to narrow the wavelength region in order to make the separation of the half-widths of the Raman spectra clearer.

(2)    X-ray studies (lines 204-215) should be significantly expanded. First of all, it is necessary to specify the wavelength of X-ray radiation used. It is necessary to compare the rocking curves with the theoretical value of the diamond rocking curve for reflection (400). The rocking curve can be calculated in kinematic approximation. It is advisable to present all the rocking curves (experimental and calculated) in one figure 8 for comparison. It is also necessary to determine the parameters of the interplanar spacing on the basis of the rocking curve and compare for different concentrations of methane.

(3)    The conducted studies do not provide an answer to the perfection of the structure of the grown films. It is necessary to use the method of X-ray topography to determine the main types of defects (dislocations, inclusions, growth bands, zonal structure), which unambiguously allows to identify the defects of crystal structure. And if possible, it is interesting to use X-ray tomography to compare the grown diamond films at different methane concentrations.

September 25, 2022

Author Response

Dear Reviewer,

Thank you for your valuable feedback, which have improved the quality of our article. Below, we provide responses to the comments raised regarding our work. A marked-up manuscript’s version showing the changes made is also provided. We hope the manuscript to attain to the standard of Materials.

Best regards,

The authors

Reviewer 2 Report

The concentration of CH4 surely affects the thickness of grown films why are you correlating FWHM. 6% to 12% why not more than that 

In our experiments, no intentional nitrogen was added to the process but it was present in feed gases as a contaminant impurity and also introduced through leakages in the vacuum system. how you can correlate growth rate to nitrogen addition?

The role of nitrogen doping needs complete mechanistic study 

page 11 line 244 

Author Response

(The authors gave the same response as above.)

Round 2

Reviewer 1 Report

The authors have made the necessary changes in the content of the article. The article can be published in present form.